ᵃ | **Open Peer Review** | Bacteriology | Research Article

# Ecological dynamics of three persistent opportunistic pathogens in hospital sinks and their potential antagonistic bacteria

Thibault Bourdin,[1] Mylène C. Trottier,[1] Marie-Ève Benoit,[2] Michèle Prévost,[3] Caroline Quach,[2] Alizée Monnier,[1] Dominique Charron,[3] Eric Déziel,[1] Philippe Constant,[1] Emilie Bédard[3]

**ABSTRACT** Sinks contaminated with opportunistic pathogens are a source of hospital-acquired infections, responsible for morbidity and mortality in neonatal intensive care units (NICUs). Understanding pathogen behavior in sinks is essential for preventing their spread. Only a few studies have examined how sink environments affect pathogen distribution through changes in drain microbiota. This research uses an integrative approach to study three major bacterial pathogens: *Pseudomonas aeruginosa*, *Stenotrophomonas maltophilia*, and *Serratia marcescens*. Sink drains in two NICUs were sampled during 2-month and 5-month periods. The diversity and abundance of opportunistic pathogens were determined at the genotypic level. Their occurrence was analyzed considering microbial communities, water parameters, faucet design, and sink usage. *P. aeruginosa*, *S. marcescens*, and *S. maltophilia* were found in 47%, 39%, and 67% of drain samples, respectively. Low genotype diversity was observed within sinks, with 1–3 genotypes per species/sample. Dominant genotypes persisted throughout the sampling periods, showing the persistence of opportunistic pathogen strains in drains. Quantification of the studied bacterial sequence types ranged from $10^3$ to $10^7$ DNA copies/mL. The heterogeneous spatial distribution of the three species between individual sink drains was primarily attributed to changes in community composition, chlorine concentrations, and faucet design. We isolated a strain of *Delftia tsuruhatensis* (Dt1S33), whose presence in the sink environment was negatively correlated with the three opportunistic pathogens. Dt1S33 reduced the capacity of the pathogens to form biofilms in laboratory co-cultures. These findings underscore the key roles of biotic and abiotic factors in the colonization of sink drains by pathogens.

**IMPORTANCE** Hospital sinks are critical reservoirs for opportunistic pathogens (OPs), increasing the risk of healthcare-associated infections, especially in vulnerable populations such as neonatal intensive care unit (NICU) patients. Our study found that 39%–67% of sink drains were persistently colonized by *Pseudomonas aeruginosa*, *Serratia marcescens*, and *Stenotrophomonas maltophilia*, with a limited number of genotypes dominating for months. Colonization patterns in drains varied between NICUs, mainly influenced by microbial community composition and sink design. Notably, *Delftia tsuruhatensis* presence was negatively correlated with OP colonization and inhibited OP biofilm formation *in vitro*. These results highlight the interplay of abiotic and biotic factors in sink colonization and suggest that antagonistic bacteria could help reduce pathogen persistence. Understanding these dynamics is crucial for developing targeted interventions to mitigate infection risks in high-risk hospital settings.

**KEYWORDS** high-throughput short sequence typing (HiSST), P-traps, drains, neonatal intensive care units (NICU), healthcare-associated infections (HAIs), opportunistic premise plumbing pathogens, *Serratia marcescens*, *Pseudomonas aeruginosa*, *Stenotrophomonas maltophilia*, *Delftia tsuruhatensis/lacustris*

**Peer Reviewers** Innocent Afeke, University of Health and Allied Sciences, Ho, Volta Region, Ghana; M. Jahangir Alam, University of Houston, Houston, Texas, USA

Address correspondence to Emilie Bédard, e.bedard@polymtl.ca.

The authors declare no conflict of interest.

See the funding table on p. 13.

Healthcare-associated infections (HAIs) represent an important public health issue, imposing considerable burdens and high costs on healthcare systems. Up to 30% of HAIs occur in intensive care units, leading to a 52% increase in mortality for infected patients (1). From 2010 to 2016, approximately 12% of HAIs occurred in neonatal intensive care units (NICUs) in Canada (2).

The conditions prevailing in the sink environment provide an ideal ecological niche for the proliferation of gram-negative opportunistic pathogens (OPs) (3). Among the potential OPs, *Pseudomonas aeruginosa*, *Serratia marcescens*, and *Stenotrophomonas maltophilia* are particularly concerning due to their intrinsic resistance to antibiotics and their frequent association with HAIs (4–9).

The bacterial species *P. aeruginosa* (*Pseudomonadaceae*), *S. marcescens* (*Yersiniaceae*), and *S. maltophilia* (*Xanthomonadaceae*) are ubiquitous in environments impacted by human activities (10) or in natural environments and various hosts (11–13). These species thrive in moist conditions (14) and exhibit metabolic versatility, allowing them to acclimatize to various ecological niches, including medical facilities and patient-related equipment (4, 6, 15–18). Their ability to form biofilms, often regulated by quorum-sensing signaling, enables them to adhere and grow on diverse surfaces, acting as significant virulence factors and enhancing their resilience (19–23). These bacteria are notable for their tolerance to antimicrobial agents and their role as OPs (24–28), responsible for a wide spectrum of human organ infections (29). Outbreaks of HAIs caused by multidrug-resistant strains, particularly for *P. aeruginosa* and *S. marcescens*, have been repeatedly reported in NICUs (9, 30–33), leading to significant morbidity and mortality in very preterm infants. Recent research reports 2.3 late-onset *Serratia* infections per 1,000 very preterm infants (34). *S. marcescens* ranks as the second most cited species in studies linking sink-related bacterial HAIs in NICUs, after *P. aeruginosa* (8).

While a wide range of opportunistic and pathogenic microorganisms have been reported in hospital environments (35–37), these three bacterial species are consistently and abundantly detected in sink drain biofilms and building water systems (26, 38). In addition, OPs may be present in a viable but non-culturable state in high concentration within biofilms, promoting persistence in such environments (39). Although previous studies have shown that hospital sinks can harbor persistent OP populations (15, 40), most work has focused on species-level detection or culture-based methods. However, the influence of environmental conditions and resident microbiota on the distribution of OP genotypes in NICU drains remains poorly understood.

Here, we address these gaps by examining whether the presence of three clinically relevant OPs (*P. aeruginosa*, *S. marcescens*, and *S. maltophilia*) in NICU sink drains is associated with environmental conditions and co-occurring microbial communities. To our knowledge, this is the first study to characterize their distribution at the genotype level across NICU drains. We also assessed the potential role of resident taxa in modulating OP occurrence, including the association of *Delftia tsuruhatensis* Dt1S33 with reduced OP colonization. The role of Dt1S33 in preventing OP colonization in sink drains is supported by bacterial community profiling and co-culture assays.

## MATERIALS AND METHODS

This study was conducted in two NICUs: NICU#1 was built in 2015 and has a capacity of 80 beds with mainly single rooms and five double rooms for twins, while NICU#2 was built before the 2000s. The layout of NICU#2 is different, with all patients in one large room of 16 beds. Sinks in NICU#1 and NICU#2 are identified by the prefixes "1" and "2," respectively. Detailed materials and methods are available as Supplemental Methods. Faucet samples (tap water and aerator biofilms) were excluded from the analysis due to their sporadic contamination, which appeared to be more attributable to stochastic events rather than systematic factors.

## Sink drain sampling and processing

In NICU#1, sampling was done every 2 weeks for 6 weeks (three sampling per sink) between January and February 2020. In NICU#2, five sampling campaigns were conducted during 5 months, between August and December 2020. All sinks were sampled in a random order at each sampling date.

The comprehensive sampling procedures are detailed in a previous study (9). Samples were processed within 6 h. In NICU#2, drain biofilm swabs and drain water were processed together and treated as a single "biofilm suspension." This decision was based on observations from the initial sampling campaign in NICU#1, in which biofilm and water samples showed highly comparable pathogen detection profiles. Moreover, the swabbing procedure inherently immerses the swab in drain water, resulting in cross-exchange between the two matrices. Combining the fractions, therefore, provides a more integrated representation of the drain environment while also streamlining sample processing without compromising the integrity of microbial detection. Samples were tested for the presence of three OPs: *P. aeruginosa*, *S. marcescens*, and *S. maltophilia*. Direct analysis of environmental genomic DNA (eDNA) was performed for drain water (WD) and drain biofilm (BD) samples.

Tap water physicochemical parameters were monitored during each sampling, with measurements of flow rate, temperature (Omega Engineering, Norwalk, CT, USA), pH, conductivity, dissolved oxygen (HQ4300 Portable Multi-Meter, HACH, London, ON, Canada), as well as total and residual chlorine (pocket colorimeter from HACH, London, ON, Canada). Drain water turbidity was also measured (HACH, London, ON, Canada).

## Sink usage frequency estimation

Sink usage frequencies were estimated by measuring temperature fluctuations every 30 s at the hot and cold tap water inlets, using temperature surface sensors (Surface Thermocouple with Self-Adhesive Backing, Omega, St-Eustache, QC, Canada) connected to data loggers (Portable Thermometer Thermocouple Data Loggers with SD Card, Omega, St-Eustache, QC, Canada). Data analyses are described in Supplemental Methods.

## Detection and genotyping of the three OPs

Isolates were analyzed by PCR targeting the three or four loci of corresponding high-throughput short sequence typing (HiSST) schemes (41, 42), equivalent to a pulsed-field gel electrophoresis (PFGE) discriminatory method. Occurrence of each OP in eDNA samples was primarily screened by PCR, targeting only one discriminating locus of the corresponding HiSST schemes to reduce manipulation effort, *bssA* for *S. marcescens* detection, *pheT* for *P. aeruginosa*, and *glnG* for *S. maltophilia* (9, 42). Library pools were sequenced at the Centre d'expertise et de services Génome Québec (Montréal, Canada), using the Illumina MiSeq PE-250 platform. The entire raw sequencing reads processing was performed using the DADA2 pipeline (43) adapted for HiSST schemes (42), using the R script "Script_RUN_FunHiSSTDada2.R" and "FunHiSSTDada2.R" function available on the dedicated GitHub repository (https://github.com/LaboPC/HiSST-schemes_TB). Barcoded primers used for library preparation and the proportion of reads remaining after each step of the DADA2 pipeline are provided in Tables S1 to S4.

## Diversity analyses

The taxonomic composition of sink microbiota was monitored through the analysis of eDNA extracts. The hypervariable regions V3–V4 (of 465 pb) of the 16S ribosomal RNA gene were amplified by PCR with primers Bakt_341F: 5′-CCTACGGGNGGCWGCAG-3′ and Bakt_805R: 5′-GACTACHVGGGTATCTAATCC-3′ (44, 45). Equimolar mixtures of PCR amplicons were sent for sequencing by Illumina MiSeq PE-250. We targeted 30,000–40,000 reads per sample. Quality control, read pair assembly, and chimera removal were performed using the DADA2 pipeline (43) on RStudio environment. Filtered sequences displaying 100% identity were gathered into amplicon sequence variant (ASV). A

representative sequence from each ASV was then compared with reference sequences from the SILVA rRNA database project (46). The final product is a list of ASVs detected, their taxonomy, and their distribution (relative abundances) in each sample (Table S1).

Before diversity analyses, ASV tables were filtered to remove sink sites with zero counts and ASVs corresponding to the three target OPs. Community data were then normalized using a Hellinger transformation with the decostand function (vegan v2.6-4), which corrects for differences in sequencing depth and reduces the influence of double zeros. Environmental variables were standardized (*z*-score) prior to constrained ordination, and highly collinear parameters were removed based on pairwise correlations and variance inflation factor (VIF) scores >5. Normalized community matrices were subsequently used for Bray-Curtis distance-based redundancy analyses.

## Quantification of the three OPs

Droplet Digital PCR (ddPCR) was used to estimate the average abundance of OPs in sink drains over the sampling period. Three distinct sampling dates were screened within both NICUs. This encompassed all the sampling dates for the NICU#1 campaign, as well as a subset of three out of five sampling dates for NICU#2. Simplex PCR was conducted for *P. aeruginosa* quantification using primers designed to target the *pheT* locus, each at a final concentration of 100 nM. Duplex PCR was optimized to achieve simultaneous quantification of both *S. marcescens* and *S. maltophilia*, targeting the *bssA* locus (primers at 100 nM each) and the *glnG* locus (primers at 250 nM each), respectively. The detailed procedures for ddPCR preparations and PCR conditions are presented in Supplemental Methods and Table S5.

## Statistical analyses

Statistical analyses were conducted within the R environment (47, 48) to explore the variables explaining the presence of OPs and the microbiota composition of sink drains. First, the number of species and their relative abundance were estimated by calculating various diversity estimators and indices (49), including species richness and the Shannon (50) and Simpson indices (51), using the vegan (v2.6-4) and dplyr (v1.4.4) packages (52, 53).

A comprehensive evaluation of how variables influenced the presence and concentration of OPs was conducted through a sequence of multivariate analyses involving the ASV matrix and physicochemical parameters. These analyses encompassed permutation analysis of variance, generalized mixed models, canonical redundancy analysis, and partitioning of variation (54). For multivariate analysis purposes, sinks showing a positive result for an OP on a single date out of three or five sampling dates for NICU#1 and NICU#2, respectively, were classified as overall negative for OPs. This classification was made to avoid false-positive bias and was based on the assumption that such occurrences represented sporadic contamination rather than true OP colonization.

## Isolation and *in vitro* tests of a potential bacterial antagonist against OPs

To verify experimentally an antagonist activity against OPs, we retrieved the bacterial strain corresponding to the ASV-1 of the sink drain 1-S33. MacConkey agar plates were inoculated with sink drain sample 1-S33 and incubated for 72 h at room temperature (22°C). A single colony morphotype was prevalent. Colonies were purified on trypticase soy agar at 22°C for 72 h. A single colony of an axenic culture was inoculated in 3 mL trypticase soy broth (TSB) for 72 h at 22°C under agitation. DNA was extracted using the procedure previously described (9). The corresponding strain was named Dt1S33.

To assess the prevalence of OPs in antagonism assays, bacterial strains were chromosomally tagged using Tn7-*lux* or Tn7-*gfp* elements, as described in Supplemental Methods. The tagged strains for *P. aeruginosa* PA14 and *S. maltophilia* 810-2 are referred to as PA14-*lux* or SM810-GFP, respectively. As chromosomal tagging of *S. marcescens* was unsuccessful, we excluded this species from the experiments.

The antagonistic potential of Dt1S33 toward strains PA14-*lux* and SM810-GFP was investigated in the context of biofilm formation. For *P. aeruginosa*, bacterial suspensions were prepared with Dt1S33 to PA14-*lux* ratios of 1:1, 2:1, and 4:1. The same method was applied for *S. maltophilia* with small modifications, as described in Supplemental Methods. Pathogenic bacteria were inoculated in 96-well plates at the same concentration under all conditions, while the Dt1S33 inoculum was adjusted based on the ratios. Following a 24 h incubation at 22°C, the 96-well plate was washed with the culture medium (10% TSB medium containing 0.5% casamino acids for PA14-*lux*, or LB medium for SM810-GFP) to remove planktonic bacteria, and fresh medium was added to each well. The overall presence of PA14-*lux* or SM810-GFP in biofilms was quantified using a Cytation3 multimode microplate reader (BioTek, Winooski, VT, USA). To validate the specificity against OP strains, the procedure was replicated by substituting *D. tsuruhatensis* Dt1S33 with the negative control strain *Burkholderia cenocepacia* K56-2 (SAMN14693155). This strain belongs to the same order (Burkholderiales) as *Delftia*.

## RESULTS

### OPs colonize most sink drains over time

Since understanding the dynamics of OPs may reveal factors influencing their presence in the sink environment of NICUs, we investigated the temporal and spatial genotypic distribution of three OP species in sink drains.

All sinks in both NICUs tested positive for at least one of the three monitored OPs, except for sink 2-S23 in NICU#2, which was rarely used, and sink 1-S33 in NICU#1, for which no immediate explanation was available. On average, more than half (51%) of sink drains were colonized by at least one OP species at any given time (Table 1). The incidence of positive drain samples remained relatively consistent over time in both NICUs, regardless of the specific pathogen involved (Fig. S1).

Close or even identical HiSST genotypic profiles were identified within the same drains over consecutive weeks (Fig. 1). The diversity of OPs per sink drain was modest in both NICUs, with an average of two distinct genotypes per sample (Tables S2 to S4). Among these, *S. marcescens* exhibited the lowest genotypic diversity within sinks, typically displaying a single dominant genotype and a maximum of four distinct short sequence types (SSTs, refers to a single locus of a HiSST scheme), varying based on the sample and targeted locus. In contrast, *S. maltophilia* displayed the highest genotypic diversity in sinks, with 4–10 unique SSTs depending on the NICU and targeted locus. Remarkably, all sink drains in NICU#2 were colonized by the same *S. marcescens* genotype, except for the OP-negative sink 2-S23. Other genotypes (showing three distinct SSTs of the *gabR* locus) were detected only once during the sampling campaign, alongside the dominant genotype in one drain sample (2-S28; Fig. 1E). These sporadic genotypes were then no longer found, while the dominant genotype remained detectable in subsequent sampling. The dominant *S. marcescens* genotype was isolated from the drain 2-S29 (strain BWD29-0280-Sm1), genotyped using HiSST analysis, and confirmed by WGS.

### NICU sink drains are colonized by a wide genotypic diversity of three OPs

Having established that diversity was very low within distinct drains, we then investigated the sink-to-sink genotypic distribution of *P. aeruginosa*, *S. marcescens*, and *S. maltophilia* in drain samples across the two NICU environments (Fig. 2). Globally, the spatial

**TABLE 1** High prevalence of three OPs in drains within two NICUs

| NICU | Average frequency (%) of positive sink drains[a] | | |
|---|---|---|---|
| | *P. aeruginosa* | *S. marcescens* | *S. maltophilia* |
| NICU#1 (*n* = 20 sinks; three sampling dates) | 35 | 40 | 68 |
| NICU#2 (*n* = 8 sinks; five sampling dates) | 78 | 35 | 63 |

[a]Determined by PCR HiSST, based on data presented in Fig. S1.

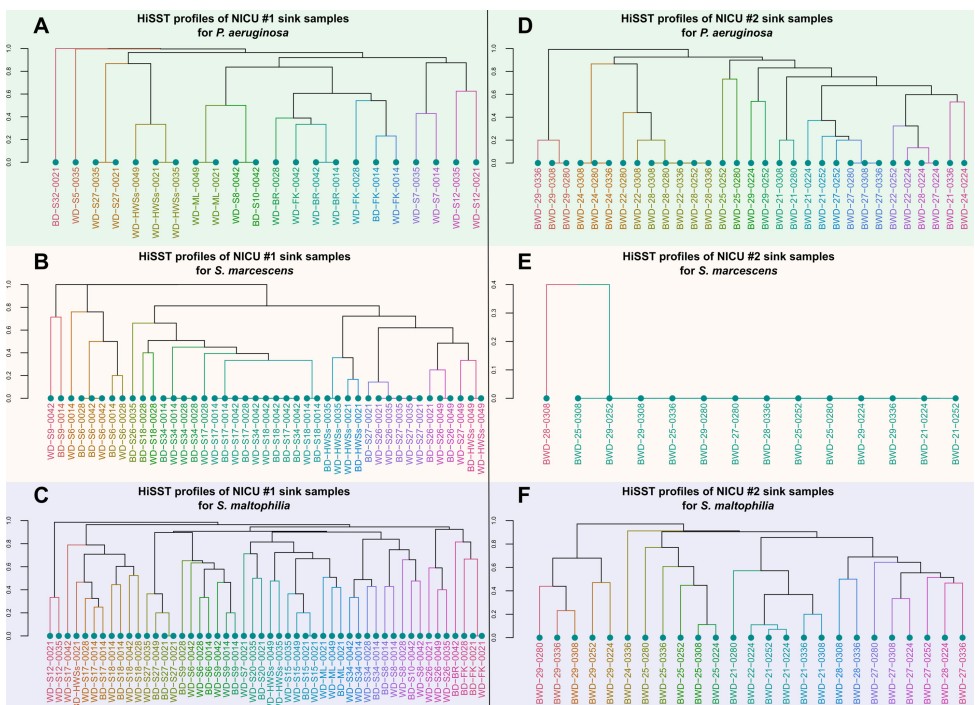

**FIG 1** Genotype comparison of *P. aeruginosa*, *S. marcescens*, and *S. maltophilia* shows high diversity in the eDNA samples retrieved from sink drains. The unweighted pair group method with arithmetic mean (UPGMA) dendrograms are based on Jaccard distance, calculated with HiSST profiles of sink samples positive from both NICUs and for the corresponding OPs. The panels on the left (A, B, and C) represent dendrograms of the samples from NICU#1, and those on the right (D, E, and F) represent dendrograms of the samples from NICU#2. Dendrograms in panels A and D correspond to *P. aeruginosa*-positive samples; dendrograms in panels B and E correspond to *S. marcescens*-positive samples; and dendrograms in panels C and F correspond to *S. maltophilia*-positive samples. A higher degree of similarity in the HiSST profiles across samples corresponds to a higher probability of colonization by a clonal strain. Samples are named based on sample type (BD for biofilm, WD for drain water, and BWD for pooled biofilm and drain water samples), followed by the sink identifier and sampling day (preceded by "0" for 2020).

distribution pattern of OPs exhibited heterogeneity among sinks, reflecting the presence of different strains between sinks in the same NICU. Notably, in two instances within NICU#1, HiSST profiles of sink pairs located in different rooms but sharing a common drain exhibited closely related genotypes (sinks with similar colors in Fig. 2). Specifically, sinks 1-S26 and 1-S27 were colonized by closely related genotypes of *S. marcescens*, and sinks 1-S17 and 1-S18 showed closely related genotypes of both *S. marcescens* and *S. maltophilia* (Fig. 1B and C). Occasionally, the colonization of two sinks by the same genotype was more likely due to their close proximity (e.g., 1-S8 and 1-S10) rather than a shared drainage system (e.g., 1-S7 and 1-S8). While in another situation, the genetic proximity observed between *P. aeruginosa* eDNA, retrieved from sinks 1-FK and 1-BR, could be associated primarily with external users (e.g., visitors) rather than the NICU healthcare workers. In NICU#2, OP-positive sinks were concentrated within a limited area, most of them in the same room, which may explain the closer HiSST profiles between sinks compared to NICU#1 (Fig. 2A and B). Importantly, handwashing stations in both NICUs exhibited the highest colonization rates for all three OPs, with over $10^5$ gene copies for each OP per mL of biofilm suspension sampled from sinks 1-HWSs and 2-S29. Sinks were generally colonized by OPs at levels ranging from $10^3$ to $10^5$ gene copies per mL, with peaks at $10^7$ gene copies per mL (Fig. 2C and D). Of note, SSTs identified in sinks located at NICU entrances were commonly present in the remaining sinks of the NICU (Tables S2 to S4).

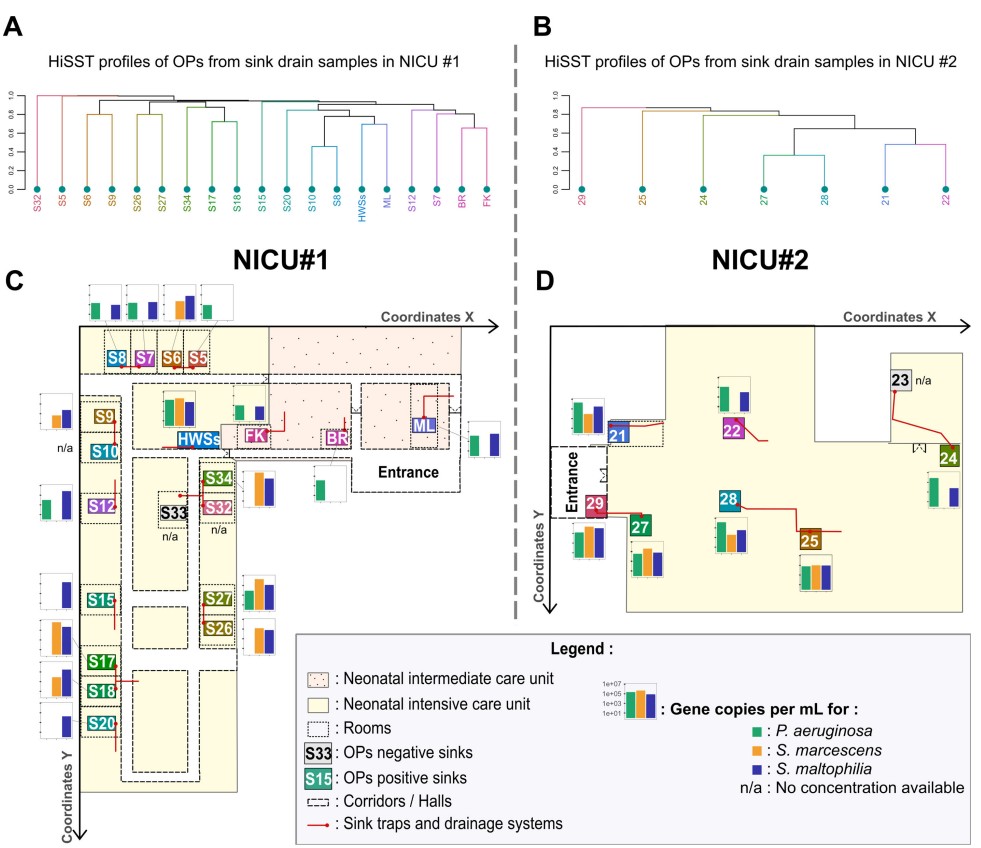

**FIG 2** The spatial distribution of HiSST profiles was heterogeneous for *P. aeruginosa*, *S. marcescens*, and *S. maltophilia* combined in both NICUs. (A, B) UPGMA dendrograms, based on Jaccard distance, utilize HiSST profiles from all OP positive drain samples, combining sequences of the three studied species. Sink colors closer together on the dendrogram suggest a higher likelihood of identical genotype colonization. Floor plans of NICU#1 (C) and NICU#2 (D) are shown. Sinks under investigation are represented by rectangles: gray sinks labeled in black indicate drain samples negative for OPs, while colored sinks labeled in white signify drains positive for at least one studied OP. Each sink color corresponds to a unique HiSST profile, illustrated as a dendrogram (shown in panel A or panel B). Histograms adjacent to sinks represent the average abundance of OPs measured in drain samples, expressed as gene copies per milliliter of filtered water (combined drain biofilm and water). Red lines symbolize gray water drainage systems, with red dots representing sampled P-traps. Coordinates *X* and *Y* were arbitrarily chosen to compare relative distances between sinks in the multivariate analysis. HWSs, intensive care handwashing station; FK, family kitchen; BR, breastfeeding room; ML, milk laboratory.

## The prevalence of all three OPs is driven by a few bacterial taxa

There was a strong association between OP occurrence and sink bacterial communities (Fig. 3; Table S6). Some abiotic variables were excluded from the redundancy analysis due to their significant collinearity with other parameters (e.g., pH and residual chlorine concentrations). In the case of NICU#2, water temperature was considered but omitted from the redundancy analysis as it covaried strongly with other parameters (hot and cold water inlet separation, pH, conductivity, dissolved oxygen; Fig. S3). The relative spatial position of sinks within the two NICUs was defined by *X* and *Y* coordinates along empirical axes defined across the distribution of sinks (Fig. 2C and D).

The structural composition of microbial communities within sink drains highlighted distinctions between patient rooms and handwashing sinks located in communal areas (Fig. S2). Notably, the predominant ASV in patient sink drains from NICU#1 was identified as *Delftia* sp., while this genus was less abundant across the four sink drains within the communal area (sinks 1-FK, 1-ML, 1-HWSs, and 1-BR in Fig. S2). In NICU#2, a notable similarity in microbiota composition was observed in two centrally located sinks

exclusively utilized by healthcare personnel (sinks 2-S22 and 2-S28), dominated by the *Elizabethkingia* genus (Fig. S2).

The factors influencing bacterial community composition differed between the two NICUs studied. In NICU#1, two key factors significantly explained variations in bacterial community composition within sink drains: residual chlorine concentrations ($P <$ 0.01) and the *X*-coordinate of sink positions ($P < 0.01$). The *X*-coordinates help to distinguish between sinks in the intensive care area and those in the intermediate care unit. In NICU#2, variations in bacterial communities were mainly explained by the presence/absence of a faucet mixing chamber ($P < 0.05$), chlorine concentration ($P < 0.05$), and pH ($P < 0.01$; collinear with chlorine residual concentrations).

Factors influencing OP presence/absence were also investigated in both NICUs, including tap water physicochemical parameters, sink localization, design, and usage frequency, and bacterial diversity (Table S6). In NICU#1, bacterial communities explain 22% of the variance in OP occurrence ($P < 0.01$), with 13% attributed jointly to pH and residual chlorine parameters, and 8% to the horizontal "*X*" axis of the unit. These three physicochemical parameters explain 17% of total variance ($P < 0.05$). A higher faucet flow rate was associated with a greater likelihood of the presence of at least one OP ($P < 0.05$). Importantly, three bacterial genera (*Enhydrobacter* sp., *Delftia* sp., and *Achromobacter* sp.) exhibited negative correlations with OP occurrence in NICU#1, explaining 22% of the OP variance. Notably, *Delftia* sp. showed the strongest negative correlation with OP presence. A fourth taxon, belonging to the genus *Elizabethkingia*, exhibited a positive covariation with OP presence. Within NICU#2 sink drains, the composition of bacterial communities independently explained 49% of the variation in OP occurrence ($P < 0.01$). Two bacterial genera, *Citrobacter* sp. and *Sphingomonas* sp., exhibited positive and negative associations with OP presence, respectively, contributing to 30% of the observed variability ($P = 0.05$), and explaining 25% of the variation jointly with residual chlorine concentrations.

Indeed, factors that impact the presence or abundance of OP species differed between NICUs (Fig. 3). A higher faucet flow rate was correlated with the presence of *S. maltophilia* and higher abundance of *S. marcescens* in drains ($P < 0.05$, Fig. S3). The separation of hot and cold water at faucet outlets (i.e., absence of a mixing chamber

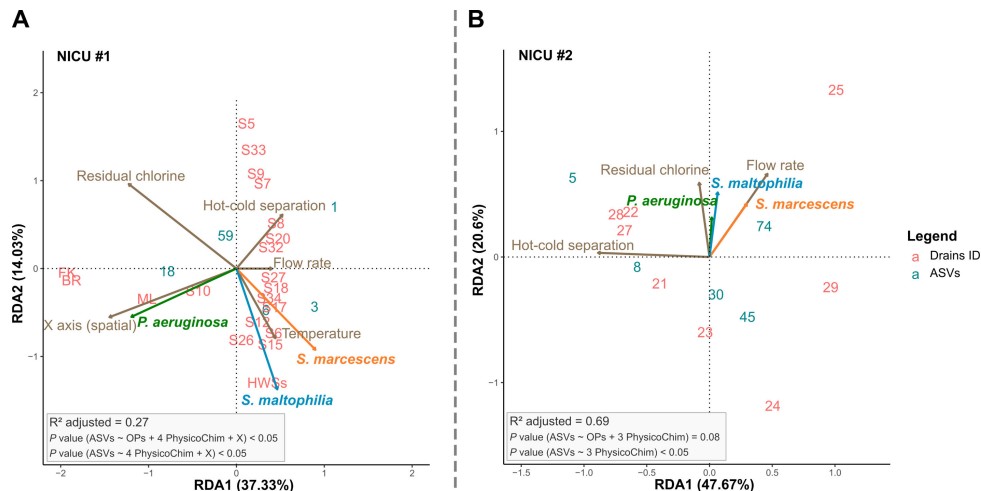

**FIG 3** Parsimonious distance-based redundancy analysis of microbial community structures colonizing NICU drains. Redundancy analysis, based on Bray-Curtis distance, assesses the relationship between bacterial community abundance, the presence of three investigated OPs, and abiotic sink parameters of NICU#1 (A) and NICU#2 (B). Each pink number on the graph corresponds to a sink drain sample, positioned according to the associated bacterial communities. Turquoise numbers represent individual amplicon sequence variants, with centroids excluded for clarity. Colored arrows represent variations in the presence/absence of OPs while brown arrows represent abiotic parameters (only non-collinear parameters explaining the most variation in communities are shown).

to blend hot and cold-water streams before outlet) exhibited a negative correlation with OP occurrence (Fig. 3) and correlated with reduced OP concentrations, as well as lower species diversity (Shannon index, $P < 0.05$) of bacterial communities in drains (Fig. S3). Lastly, *S. marcescens* and *S. maltophilia* were both correlated with similar variations in bacterial communities' composition. On the other hand, *P. aeruginosa* showed no correlation with the variation in the compositional structure of bacterial communities but was associated with greater bacterial diversity in drains (Richness indicator, $P < 0.01$).

Unexpectedly, the spatial arrangement of sinks—whether they shared the same drains directly or not (each instance of shared drains being identified using a unique ID)—had no impact on the prevalence of OP in sink drains of both NICUs.

### *D. tsuruhatensis*, a potential antagonist against OPs

The ASV assigned to *Delftia* sp. (ASV-1) was negatively correlated with the presence of OPs in NICU#1 drains (Fig. S2). This correlation was particularly striking in sink drain 1-S33, in which OPs were absent while *Delftia* sp. predominated. To investigate the antagonistic properties of *Delftia* sp., strain Dt1S33 was isolated from sink drain 1-S33, and its genome was fully sequenced and annotated (Table S7). WGS identified the species as *D. tsuruhatensis* (Fig. S4). We investigated its ability to inhibit biofilm formation by two different tagged OP strains: PA14-*lux* (*P. aeruginosa*) and SM810-GFP (*S. maltophilia*). These strains were co-cultured with Dt1S33 at various ratios. As the proportion of Dt1S33 within the initial population increased, there was a significant reduction in the abundance of both OPs in the biofilm compared to the control group (Fig. 4). Our findings suggest that Dt1S33 prevents biofilm formation by *P. aeruginosa* and *S. maltophilia* in a dose-dependent manner. Under the same conditions, we found that the presence of *B. cenocepacia*, another species belonging to the Burkholderiales order, like *Delftia*, did not impact the biofilm production by *P. aeruginosa* PA14 and *S. maltophilia* 810-2 (Fig. 4). This underscores the apparent specificity of the biofilm prevention effect caused by *D. tsuruhatensis* Dt1S33.

## DISCUSSION

This study aimed to identify variables shaping the temporal and spatial dynamics of three major OPs (*P. aeruginosa*, *S. marcescens*, and *S. maltophilia*) in sinks of two NICUs, from the faucet to the P-trap.

### OPs thrive in hospital sink drains

We found a significant proportion of sink drains colonized with OPs; approximately half consistently harbored at least one of the three studied bacterial species, consistent with colonization rates reported in other studies (36, 55). *S. maltophilia* stood out as the predominant OP found in NICU sinks (followed by *P. aeruginosa* then *S. marcescens*), with about two-thirds of sinks colonized. Moreover, most sink drains tested positive at least once during our sampling period (19 out of 20 positive drains in NICU#1, 7 out of 8 in NICU#2). Overall, positive sink drains remained colonized by the same genotype over time. This is consistent with our previous longitudinal study of *S. marcescens* in NICU#1, in which a single genotype persisted in sink drains for more than 1 year (9). Together, these findings strengthen the evidence that specific OP strains can colonize and persist in hospital sink environments (15). On the other hand, sinks located in various patient rooms within the same NICU were found to be colonized by different genotypes of pathogens, underscoring the diverse origin of pathogens in the NICU environment. Another important finding was the high prevalence and abundance of OPs in sinks located at the NICU entrances, which consistently yielded the highest contamination rates. These findings suggest that new OP strains may be introduced into the NICU via the hands of visitors and personnel, as shown by the diverse HiSST profiles in entrance sinks, some shared with patient-room sinks.

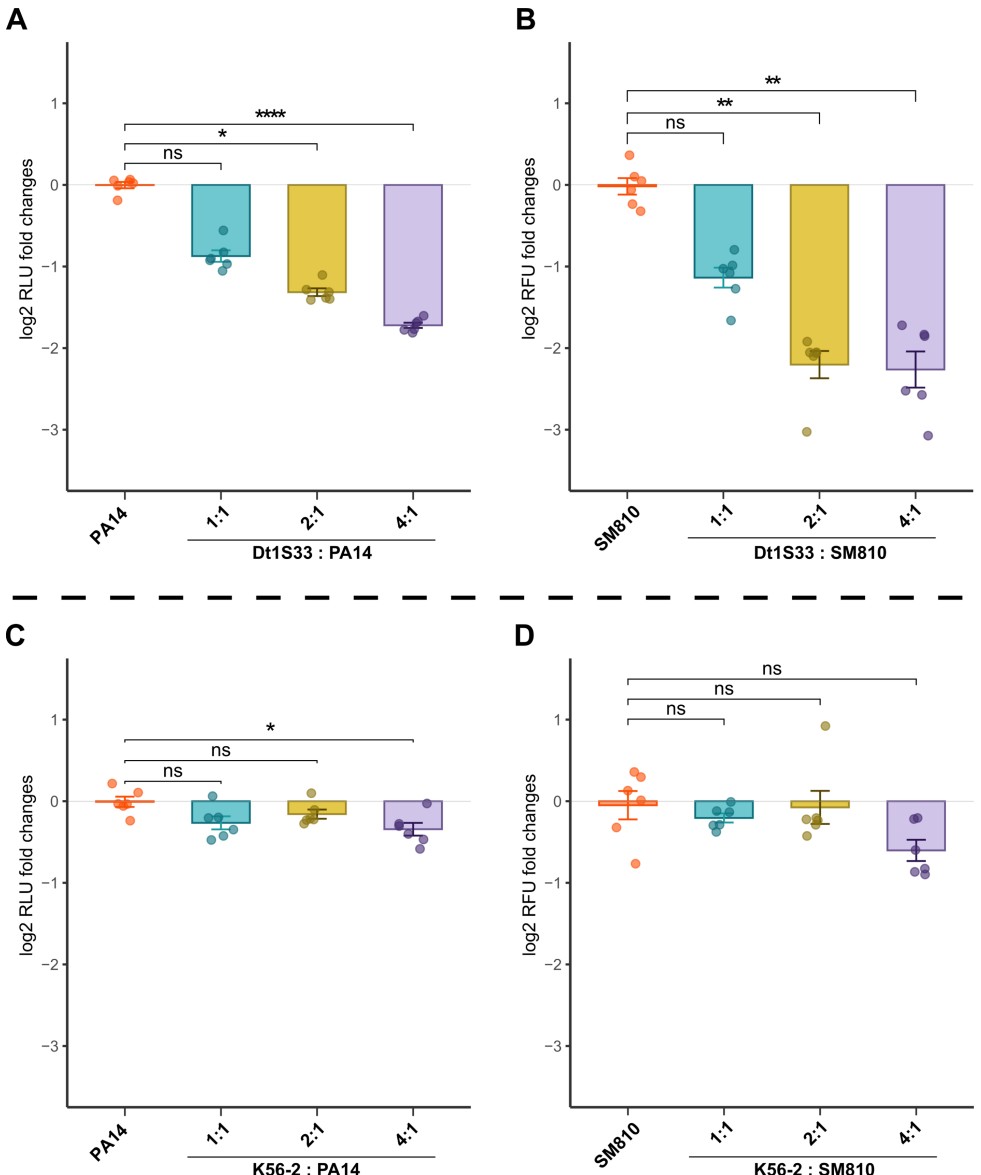

**FIG 4** Biofilm growth of two OPs in the presence of *D. tsuruhatensis* Dt1S33 or *B. cenocepacia* K56-2. Biofilms were cultivated for 24 h at 22°C in 96-well plates. (A, C) Luminescence fold changes showing the growth of *P. aeruginosa* PA14-*lux*, and (B, D) fluorescence fold changes showing the growth of *S. maltophilia* SM810-GFP in the presence of different ratios of *D. tsuruhatensis* Dt1S33 (A, B) or *B. cenocepacia* K56-2 (C, D) in the biofilm (*n* = 6). Values are compared to the growth of *P. aeruginosa* PA14-*lux* or *S. maltophilia* SM810-GFP alone. Each replicate represents an independent biofilm. Data are presented as mean log2 ± standard deviation (error bars) from six replicates. Asterisks denote *P* values for Kruskal-Wallis tests: **, *P* < 0.01; ***, *P* < 0.001; ****, *P* < 0.0001; "ns" if non-significant. RLU, relative light units; RFU, relative fluorescence units.

## Sink microbial diversity drives OP dynamics in drains

Employing multivariate analyses, we identified key factors explaining the presence of OP and the compositional variation of bacterial communities. While these analyses highlight associations, they remain hypotheses based on correlations within this complex system. The diversity of microbial communities was the main factor explaining the presence of OPs in sink drains. However, a significant portion of the variation remains unexplained and could be related to specific uses of the sink or unmeasured external factors affecting the sink environment. Among the evaluated tap water factors, residual chlorine concentration correlated with the OP variation between sink drains, although

its influence was somewhat limited. These findings align with previous studies that also reported the impact of residual chlorine on bacterial survival (56, 57). In our study, the minimal variation in pH (7.7–8.0 in NICU#1, 7.3–8.1 in NICU#2) and chlorine concentration (0.0–0.3 mg/L residual chlorine in both NICUs) made it difficult to draw conclusions on the effect of this variable on the OPs. Disparities have been noted in the factors that impact the presence or concentration of OP species. Indeed, *P. aeruginosa*, which resists chlorine at drinking water concentrations (17, 55, 58), exhibited a positive correlation with residual chlorine, whereas *S. maltophilia* displayed a negative correlation trend. The separation of hot and cold water at faucet outlets was negatively correlated with OP presence. In fact, the presence of faucet temperature mixing valve results in minimal temperature variation at faucet outlets and, consequently, in drains, and setpoint temperatures approach the optimal growth conditions for OP (typically temperate temperatures ranging between 20℃ and 38℃). Furthermore, a positive covariation was observed between higher faucet flow rate and the presence of *S. marcescens* in drains. A plausible hypothesis is that a higher flow rate may select for certain bacterial species more readily adhering to drains, as supported by biofilm adhesin genes in *S. marcescens* ED4677 (Table S7), while reducing other bacteria (59, 60).

Factors governing the presence of OPs and the composition of bacterial communities varied between the two NICUs. These observations underscore differences in environmental contexts between units, primarily originating from the widely disparate architectural layouts characterizing the two NICUs. In NICU#1, most sinks are located in separate patient rooms, with some sharing a common drain collector. The bacterial diversity variation through *X*-coordinates among sinks in NICU#1 (Fig. 2 and 3) suggests that sink position influences bacterial communities within the drain systems, possibly due to differences in usage or user behaviors. Contrasted composition of microbial communities has previously been shown in sink drains within separate patient rooms compared to those in shared spaces (61). In contrast, NICU#2 sinks are located in a single, extended area that serves multiple purposes, including handwashing at the NICU entrance, toilet handwashing, nurse rest room, patient healthcare, and lactation room. Distinctions in NICU design may contribute to variations in physicochemical parameters and bacterial community composition, as well as influence the responsiveness of the OPs to sink environmental factors (62).

Our results suggest that sink usage is the primary factor influencing microbial diversity, followed by sink design (i.e., sink location, tap water flow rate, and faucet mixing chamber) and tap water properties, which in turn affect the colonization of OPs in drains. Therefore, the role of microbial diversity appears to be a promising indicator of the external factors likely to influence OP colonization in sink drains. Although our findings have limitations (see Supplemental Methods), they suggest the important role of microbial interactions in controlling the proliferation of OPs, which is consistent with previous findings (63).

## Individual OPs exhibit distinct distribution patterns

Considering all the covariations observed above, *S. marcescens* and *S. maltophilia* seemed to share similar ecological niches. Moreover, handwashing sinks were exposed to higher bacterial colonization rates and to greater external influx of microorganisms, in contrast to sinks located in patients' rooms. Combining this result with its distribution along the *X*-axis of NICU#1, it can be inferred that sinks at the entrance of NICUs were more likely to be colonized by *P. aeruginosa*, possibly due to regular input from outside the NICUs.

Strikingly, all contaminated drains in NICU#2 were colonized by a single *S. marcescens* genotype, with transient introductions of others failing to persist. This suggests strong ecological barriers to colonization, likely mediated by microbial competition or strain-specific adaptation (not directly measured). Unlike NICU#1, where multiple genotypes and recurrent patient infections have been documented (9), NICU#2 showed neither infection cases nor genotype diversity. The dominant strain may be highly adapted to its niche, potentially through copper tolerance genes enabling survival in copper plumbing

(Table S7). The absence of clinical cases may, in turn, contribute to the limited diversity observed in sink-associated *S. marcescens*. Differences between the NICUs likely reflect infrastructure, with construction age and plumbing materials shaping metal exposure and, consequently, microbial dynamics.

## *D. tsuruhatensis*, a promising OP antagonist

Strain *D. tsuruhatensis* Dt1S33 predominated in a sink consistently negative for OPs and exhibited the strongest inverse correlation with the three OPs in other sinks of the same NICU, as evidenced by redundancy analysis. More precisely, *D. tsuruhatensis* (or *D. lacustris*, as both ultimately belong to the same species) is a member of the *Comamonadaceae* family and is frequently isolated from soil, water, and plant sources (64). *D. tsuruhatensis* is an environmental species, prevalent in sink drains (Fig. S2) (62), and seldom conclusively associated with opportunistic infections (65, 66). However, conclusive validation of its opportunistic pathogenic nature is hindered by limited supporting data and challenges in accurate taxonomic identification. Intriguingly, existing literature documents the antagonistic properties exhibited by certain strains of *Delftia* spp., such as *D. tsuruhatensis* (67)/*D. lacustris* (68), against various plant pathogens. Studies suggest that extracts of *D. tsuruhatensis* inhibit quorum sensing and biofilm formation of *P. aeruginosa* (69, 70). Furthermore, *D. tsuruhatensis* has demonstrated antimicrobial activity against clinically relevant multidrug-resistant pathogens (71).

Strain Dt1S33 thrives at room temperature, displaying a higher growth rate than the OPs studied, as observed during preculture preparation (data not shown), although growth is limited above 30°C. This mesophilic preference confers a strategic advantage against OPs, ensuring the strain's effective establishment in sink drains where water temperature is approximately the same as room temperature. Thus, the optimal development and biofilm formation of *D. tsuruhatensis* in sink drains could potentially suggest competitive exclusion of OPs by this bacterium.

This strain is susceptible to most antibiotics (Table S7), except for its intrinsic resistance to aminoglycosides. No acquired resistance genes were predicted using the ResFinder 4.7.2 (72) and CARD (73) databases. The genomic features of Dt1S33 underscore its resilience in premise plumbing and antagonistic potential against pathogens (Table S7). Genes for copper and potassium homeostasis, along with biofilm synthesis and flagellar motility, enhance its adaptability to metal stress and environmental conditions. Antagonistic traits, including bacteriocin-related genes, coupled with cofactor biosynthesis pathways, support pathogen suppression and metabolic flexibility, reinforcing its competitive advantage in plumbing ecosystems. The capacity of *D. tsuruhatensis* to inhibit *P. aeruginosa* and *S. maltophilia* biofilm formation provides valuable insight into the potential antagonistic effects on the three monitored OPs in the sink drain. Several mechanisms could underlie this effect, including quorum-sensing inhibition as proposed in previous studies (74–76), which could limit the establishment of OP biofilms (70). However, further experiments under more complex and environmentally realistic conditions are needed to determine whether Dt1S33 exerts similar effects *in situ*. No matter the exact mechanisms, our results highlight the impact of microbial diversity composition on OPs.

## Conclusion

This comprehensive study provides new insights into the ecology and dynamics of three gram-negative OPs, within two intensive care units.

Bacterial community diversity, influenced by sink design (i.e., sink location, tap water flow rate, and faucet mixing chamber) and abiotic factors (i.e., temperature, pH, and chlorine from tap water), indirectly reflects conditions that promote pathogen colonization in drains. Our findings underscore the nuanced ecological distinctions among different pathogens, highlighting the ecological patterns that differentiate *P. aeruginosa* from *S. marcescens*. A potential antagonistic role of certain members of the drain microbiota is proposed, supporting the relevance of future studies to identify

the conditions limiting the presence of OPs. For example, a strain of *D. tsuruhatensis* exhibited a negative correlation with the three OPs in drain environments and inhibited their biofilm formation *in vitro*.

Bioaugmentation with such antagonistic bacteria, or with probiotics that promote niche competition within sinks, may support the establishment of beneficial microbial communities in drains and thereby reduce pathogen exposure and infection risk (77). However, bioaugmentation or biocontrol strategies remain highly preliminary and would require extensive safety assessments, long-term stability testing, and validation under real NICU conditions before being considered, particularly for vulnerable populations such as premature infants.

## ACKNOWLEDGMENTS

We thank Yves Fontaine (Polytechnique Montréal) for his participation in sampling, the hospital staff involved in the project, and Marie-Christine Groleau (INRS) for her assistance in developing the protocol used to conduct *in vitro* antagonism assays.

This work was supported by the Natural Sciences and Engineering Research Council of Canada and the Canadian Institutes of Health Research through the Industrial Chair on Drinking Water and the Collaborative Health Research Program funding (CHRP 523790-18). Dr. Caroline Quach is the Tier-1 Canada Research Chair (CRC-2019-00055) in Infection Prevention. Eric Déziel is the Tier-1 Canada Research Chair (CRC-2023-00119) in Fundamental and Applied Sociomicrobiology.

## AUTHOR AFFILIATIONS

[1]Centre Armand-Frappier Santé Biotechnologie, Institut national de la recherche scientifique (INRS), Laval, Quebec, Canada

[2]Centre de Recherche CHU Sainte-Justine, Université de Montréal, Montréal, Quebec, Canada

[3]Polytechnique Montréal, Montréal, Quebec, Canada

## AUTHOR ORCIDs

Thibault Bourdin  http://orcid.org/0000-0001-5257-9374
Caroline Quach  http://orcid.org/0000-0002-1170-9475
Eric Déziel  https://orcid.org/0000-0002-4609-0115
Philippe Constant  http://orcid.org/0000-0003-2739-2801
Emilie Bédard  http://orcid.org/0000-0002-1447-929X

## FUNDING

| Funder | Grant(s) | Author(s) |
| --- | --- | --- |
| Canadian Institutes of Health Research | CHRP 523790-18 | Michèle Prévost |
| | | Caroline Quach |
| | | Eric Déziel |
| Natural Sciences and Engineering Research Council of Canada | CHRP 523790-18 | Michèle Prévost |
| | | Caroline Quach |
| | | Eric Déziel |

## AUTHOR CONTRIBUTIONS

Thibault Bourdin, Conceptualization, Data curation, Formal analysis, Investigation, Methodology, Resources, Software, Validation, Visualization, Writing – original draft, Writing – review and editing | Mylène C. Trottier, Formal analysis, Investigation, Methodology, Visualization, Writing – review and editing | Marie-Ève Benoit, Data curation, Formal analysis, Investigation, Methodology, Writing – review and editing |

Michèle Prévost, Conceptualization, Funding acquisition, Investigation, Methodology, Project administration, Supervision, Validation, Writing – review and editing | Caroline Quach, Conceptualization, Funding acquisition, Investigation, Methodology, Project administration, Supervision, Validation, Writing – review and editing | Alizée Monnier, Investigation | Dominique Charron, Project administration, Writing – review and editing | Eric Déziel, Conceptualization, Data curation, Funding acquisition, Investigation, Methodology, Project administration, Resources, Supervision, Validation, Writing – review and editing | Philippe Constant, Conceptualization, Data curation, Formal analysis, Investigation, Methodology, Project administration, Resources, Supervision, Validation, Visualization, Writing – original draft, Writing – review and editing | Emilie Bédard, Conceptualization, Investigation, Methodology, Project administration, Supervision, Validation, Visualization, Writing – review and editing

## DATA AVAILABILITY

Raw sequencing reads have been deposited in the Sequence Read Archive of the NCBI under BioProject no. PRJNA1042964.

## ADDITIONAL FILES

The following material is available online.

### Supplemental Material

**Supplemental figures (mSystems01546-25-s0001.docx).** Figures S1 to S4.
**Supplemental Methods (mSystems01546-25-s0002.docx).** Details of NICU environment and sink sampling, microbial detection and genotyping, sequencing approaches, bacterial transformation and antagonism assays, data accession, and study limitations.
**Graphical abstract (mSystems01546-25-s0003.eps).** Summary.
**Table S1 (mSystems01546-25-s0004.xlsx).** Sequencing results of the 16S ribosomal RNA gene amplification.
**Table S2 (mSystems01546-25-s0005.xlsx).** Sequencing results for NICU #1 and #2, using the *P. aeruginosa* HiSST scheme and bioinformatics pipeline.
**Table S3 (mSystems01546-25-s0006.xlsx).** Sequencing results for NICU #1 and #2, using the *S. marcescens* HiSST scheme and bioinformatics pipeline.
**Table S4 (mSystems01546-25-s0007.xlsx).** Sequencing results for NICU #1 and #2, using the *S. maltophilia* HiSST scheme and bioinformatics pipeline.
**Table S5 (mSystems01546-25-s0008.xlsx).** ddPCR DNA quantification values for the three targeted bacteria in both NICUs.
**Table S6 (mSystems01546-25-s0009.xlsx).** Biotic and abiotic parameters in two neonatal intensive care units.
**Table S7 (mSystems01546-25-s0010.xlsx).** Genome annotations for Dt1S33 and ED4677.

### Open Peer Review

**PEER REVIEW HISTORY (review-history.pdf).** An accounting of the reviewer comments and feedback.

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
