## [Reviewer comments · mSystems]

Ecological dynamics of three persistent opportunistic pathogens in hospital sinks and their potential antagonistic bacteria

Thibault Bourdin, Mylène Trottier, Marie-Ève Benoit, Michèle Prévost, Caroline Quach, Alizée Monnier, Dominique Charron, Eric Déziel, Philippe Constant, and Emilie Bédard

Corresponding Author(s): Emilie Bédard, Polytechnique Montréal

Review Timeline:

Submission Date:	October 28, 2025
Editorial Decision:	December 1, 2025
Revision Received:	December 17, 2025
Accepted:	January 8, 2026

Editor: Michaeline Albright

Reviewer(s): Disclosure of reviewer identity is with reference to reviewer comments included in decision letter(s). The following individuals involved in review of your submission have agreed to reveal their identity: Innocent Afeke (Reviewer #1); M Jahangir Alam (Reviewer #2)

Transaction Report:

DOI: <https://doi.org/10.1128/msystems.01546-25>

Re: mSystems01546-25 (Ecological dynamics of three persistent opportunistic pathogens in hospital sinks and their potential antagonistic bacteria)

Dear Dr. Emilie Bédard:

Thank you for the privilege of reviewing your work. Below you will find instructions from the mSystems editorial office, and the reviewer comments.

Revision Guidelines

Sincerely,
Michaeline Albright
Editor
mSystems

Reviewer #1 (Comments for the Author):

Abstract

1. I suggest that the authors include key methodological details (e.g., sequencing/typing approach, water parameter measurements, type of quantification) to give readers sufficient context for the term "genotypic level."
2. Could the authors replace vague expressions like "over half of sinks" with precise percentages or ratios for clarity and transparency?

3. I recommend softening the claim that "no previous study has examined..." since related work exists; phrasing such as "few studies have..." would be more accurate.

Introduction

1. I suggest that the authors streamline the introduction and more clearly articulate the specific knowledge gap, distinguishing what prior sink ecology studies have done and what this study adds.
2. Could the authors revise statements that imply causation (e.g., "driven by") to reflect associations rather than confirmed mechanisms?
3. I recommend restructuring the introduction to follow a clearer flow-from clinical relevance to sink reservoirs, existing evidence, knowledge gaps, and explicit study objectives.

Methods

1. Could the authors justify the large difference in sampling duration between NICUs, as this discrepancy may bias temporal comparisons?
2. I suggest providing a stronger justification for combining biofilm and water samples in NICU#2, or presenting an analysis showing that this does not distort microbial signatures.
3. Could the authors clarify key methodological steps-such as normalization for diversity analyses, ddPCR gene targets, and assay calibration-which currently appear incomplete or relegated to the Supplementary Data?

Results

1. I suggest that claims of "persistent colonization" be moderated, given the limited number and unequal timing of sampling events across the two NICUs.
2. Could the authors clarify what is meant by "heterogeneous spatial distribution," specifying whether variability occurs between rooms, sink types, or drainage networks?
3. I recommend tempering interpretations of *Delftia*'s antagonistic role, emphasizing that correlations and in vitro inhibition do not necessarily indicate ecological suppression in sink drains.

Discussion and Conclusion

1. Could the authors distinguish more explicitly between data-driven findings and speculative interpretations, particularly regarding ecological mechanisms and infrastructure effects that were not directly measured?
2. I suggest revising discussions of niche similarity and mechanistic explanations to better align with the available evidence and avoid overstating conclusions.
3. I recommend tempering the conclusion regarding bioaugmentation, as this application is premature without safety assessments, long-term stability testing, and validation in real NICU environments.

Reviewer #2 (Comments for the Author):

This study provides a rigorous and multifaceted examination of the ecological determinants governing the persistence of *Pseudomonas aeruginosa*, *Serratia marcescens*, and *Stenotrophomonas maltophilia* in NICU sink drains. By combining longitudinal sampling, genotypic characterization, microbial community analysis, and abiotic measurements, the authors present one of the most comprehensive assessments to date of pathogenic colonization in hospital sink environments. A key strength of the work lies in its genotype-level resolution, which reveals low diversity and the long-term persistence of dominant sequence types within drains. This finding underscores the capacity of specific pathogenic lineages to stably colonize plumbing systems, potentially serving as chronic reservoirs for hospital-acquired infections. The study also demonstrates that spatial heterogeneity in pathogen distribution is driven primarily by variation in drain community composition, residual chlorine, and faucet design, illustrating how engineering and environmental parameters jointly shape microbial ecology in clinical settings. The isolation and characterization of *Delftia tsuruhatensis* Dt1S33 represent a notable contribution. Its negative correlation with opportunistic pathogens in situ, coupled with its inhibition of pathogen biofilm formation in vitro, provides compelling preliminary evidence for antagonistic interactions within sink microbiota. This insight suggests that ecological or probiotic strategies-such as bioaugmentation with antagonistic strains-may complement or enhance traditional disinfection approaches. However, the translational potential of such methods, particularly in vulnerable populations such as premature infants, requires careful evaluation. Overall, this study advances current understanding of environmental reservoirs of healthcare-associated pathogens and highlights the intertwined roles of biotic and abiotic factors in shaping sink colonization dynamics. The findings have important implications for infection control and point toward innovative, ecology-informed interventions capable of reducing pathogen persistence in high-risk hospital units.

Reviewer comments:

Reviewer #1 (Comments for the Author):

We thank the reviewer for their careful evaluation of our manuscript and for the insightful and constructive suggestions, which have helped us improve the clarity and quality of the work.

Abstract

- ***Comment 1***

I suggest that the authors include key methodological details (e.g., sequencing/typing approach, water parameter measurements, type of quantification) to give readers sufficient context for the term "genotypic level".

Answer 1

We agree that additional methodological details could provide useful context. However, the abstract is constrained by a 250-word limit imposed by the journal, and we are unable to further condense the existing content without losing essential information. The full manuscript includes a detailed description of the sequencing/typing approach, water parameter measurements, and quantification methods.

- ***Comment 2***

Could the authors replace vague expressions like "over half of sinks" with precise percentages or ratios for clarity and transparency?

Answer 2

This comment refers to the "Importance" section. We revised the phrasing from "over half of sink drains" to "39% to 67% of sink drains" (line 38), as recommended.

- ***Comment 3***

I recommend softening the claim that "no previous study has examined..." since related work exists; phrasing such as "few studies have..." would be more accurate.

Answer 3

As recommended, we revised the statement from “No previous study has examined” to “Only a few studies have examined” (line 49).

Introduction

- *Comment 4*

I suggest that the authors streamline the introduction and more clearly articulate the specific knowledge gap, distinguishing what prior sink ecology studies have done and what this study adds. Could the authors revise statements that imply causation (e.g., "driven by") to reflect associations rather than confirmed mechanisms? I recommend restructuring the introduction to follow a clearer flow—from clinical relevance to sink reservoirs, existing evidence, knowledge gaps, and explicit study objectives.

Answer 4

We reorganized the introduction to follow the requested structure: clinical relevance, OP ecology in sinks, existing evidence, the knowledge gap, and study objectives.

To clarify the specific knowledge gap, we added the following sentences (lines 94 to 102): “Although previous studies have shown that hospital sinks can harbor persistent OP populations (16,17), most work has focused on species-level detection or culture-based methods. However, the influence of environmental conditions and resident microbiota on the distribution of OP genotypes in NICU drains remains poorly understood. Here, we address these gaps by examining whether the presence of three clinically relevant OPs (*P. aeruginosa*, *S. marcescens*, and *S. maltophilia*) in NICU sink drains is associated with environmental conditions and co-occurring microbial communities. To our knowledge, this is the first study to characterize their distribution at the genotype level across NICU drains.”

Additionally, to avoid implying causation, we replaced the phrase “is driven” with “is associated” (line 100).”

Methods

- *Comment 5*

Could the authors justify the large difference in sampling duration between NICUs, as this discrepancy may bias temporal comparisons?

Answer 5

The difference in sampling duration between NICUs was due to the COVID-19 shutdown in 2020, which required us to stop sampling in NICU#1 earlier than planned. We recognize that this discrepancy could bias temporal comparisons. To minimize this effect, we analyzed NICU#1 and NICU#2 largely separately rather than pooling their data, especially given the structural differences between the two units (as detailed in the limitations section in Supplemental Methods).

Despite the unequal sampling periods, we believe the observed persistence patterns are reliable. Several genotypes were repeatedly detected over weeks to months, even in sinks exposed to intensive daily use (tens to hundreds of flushing events per day; Table S6), supporting the ability of these strains to persist in drain environments.

- **Comment 6**

I suggest providing a stronger justification for combining biofilm and water samples in NICU#2, or presenting an analysis showing that this does not distort microbial signatures.

Answer 6

Thank you for your comment. This decision was based on observations from the initial sampling campaign in NICU#1, in which biofilm and water samples showed highly comparable pathogen detection profiles. Moreover, the swabbing procedure inherently immerses the swab in drain water, resulting in unavoidable mixing and cross-exchange between the two matrices. Under these conditions, analyzing them separately would impose an artificial distinction not reflective of the sampling process. Combining the fractions therefore provides a more integrated representation of the drain environment while also streamlining sample processing without compromising the integrity of microbial detection.

We have improved our justification as follows (line ##):

“In NICU#2, drain biofilm swabs and drain water were processed together and treated as a single “biofilm suspension”. This decision was based on observations from the initial sampling campaign in NICU#1, in which biofilm and water samples showed highly comparable pathogen detection profiles. Moreover, the swabbing procedure inherently immerses the swab in drain water, resulting in cross-exchange between the two matrices. Combining the fractions therefore provides a more integrated representation of the drain environment while also streamlining sample processing without compromising the integrity of microbial detection.”

- **Comment 7**

Could the authors clarify key methodological steps-such as normalization for diversity analyses, ddPCR gene targets, and assay calibration-which currently appear incomplete or relegated to the Supplementary Data.

Answer 7

Thank you for pointing out these missing key methodological steps.

For the section “16S rRNA gene amplicon sequencing” (lines 151 to 162), we moved the whole paragraph from the supplemental data to the main text. We added a new paragraph for diversity analyses (lines 163 to 169): “Before diversity analyses, ASV tables were filtered to remove sink sites with zero counts and ASVs corresponding to the three target opportunistic pathogens. Community data were then normalized using a Hellinger transformation with the *decostand* function (*vegan* v2.6-4), which corrects for differences in sequencing depth and reduces the influence of double-zeros. Environmental variables were standardized (z-score) prior to constrained ordination, and highly collinear parameters were removed based on pairwise correlations and VIF scores > 5. Normalized community matrices were subsequently used for Bray-Curtis distance-based redundancy analyses.”

For ddPCR, we added more details on the gene targeted (lines 174 to 177):

“Simplex PCR was conducted for *P. aeruginosa* quantification using primers designed to target the *pheT* locus, each at a final concentration of 100 nM. Duplex PCR was optimized to achieve simultaneous quantification of both *S. marcescens* and *S. maltophilia*, targeting the *bssA* locus (primers at 100 nM each) and the *glnG* locus (primers at 250 nM each), respectively.”

Results

- **Comment 8**

I suggest that claims of "persistent colonization" be moderated, given the limited number and unequal timing of sampling events across the two NICUs.

Answer 8

We agree that the number and timing of sampling events between the two NICUs are unequal. However, our assertion of persistent colonization is supported not only by the repeated detection of identical genotypes over time in NICU#2, but also by prior longitudinal work in NICU#1 on

Serratia marcescens, where we documented the same genotype persisting in sink drains for over a year. We have now added a clarifying statement in the Discussion to reflect that this prior evidence supports our interpretation of long-term colonization in NICU environments (lines 348 to 351): “Overall, positive sink drains remained colonized by the same genotype over time. This is consistent with our previous longitudinal study of *S. marcescens* in NICU#1, in which a single genotype persisted in sink drains for more than one year (9). Together, these findings strengthen the evidence that specific OP strains can colonize and persist in hospital sink environments (15)”.

- **Comment 9**

Could the authors clarify what is meant by "heterogeneous spatial distribution," specifying whether variability occurs between rooms, sink types, or drainage networks?

Answer 9

We have clarified the meaning of “heterogeneous spatial distribution” by specifying that the variability refers to differences between individual sink drains. This clarification has been added in the revised manuscript (line 61).

- **Comment 10**

I recommend tempering interpretations of *Delftia*'s antagonistic role, emphasizing that correlations and *in vitro* inhibition do not necessarily indicate ecological suppression in sink drains.

Answer 10

We revised the text accordingly to clarify that while several mechanisms could underlie the observed effects, additional experiments under more complex and environmentally realistic conditions are required to determine whether Dt1S33 exerts similar effects *in situ*. Lines 444 to 447: “Several mechanisms could underlie this effect, including quorum-sensing inhibition as proposed in previous studies (74–76), which could limit the establishment of OP biofilms (70). However, further experiments under more complex and environmentally realistic conditions are needed to determine whether Dt1S33 exerts similar effects *in situ*”.

Discussion and Conclusion

- **Comment 11**

Could the authors distinguish more explicitly between data-driven findings and speculative interpretations, particularly regarding ecological mechanisms and infrastructure effects that were not directly measured? I suggest revising discussions of niche similarity and mechanistic explanations to better align with the available evidence and avoid overstating conclusions.

Answer 11

We revised the manuscript to more clearly distinguish data-driven results from our interpretation. Specifically, we explicitly note when proposed ecological mechanisms or infrastructure influences were not directly measured (line 409), and we added the statement: “While these analyses highlight associations, they remain hypotheses based on correlations within this complex system” (line 385-385), and “However, a significant portion of the variation remains unexplained and could be related to specific uses of the sink or unmeasured external factors affecting the sink environment” (lines 586 to 388). Additionally, throughout the manuscript we systematically adopted conditional language (e.g., “suggest”, “may”, “potentially”, “likely”, “A plausible hypothesis...”) to ensure that all interpretative statements are clearly identified as hypotheses rather than confirmed mechanisms.

- **Comment 12**

I recommend tempering the conclusion regarding bioaugmentation, as this application is premature without safety assessments, long-term stability testing, and validation in real NICU environments.

Answer 12

Thank you for this important point. We fully agree that these strategies must undergo careful evaluation before they are implemented. We therefore revised the text to emphasize that any such application is highly preliminary at this stage. We now explicitly stated that implementation would require rigorous safety evaluation, long-term stability studies, and testing in real NICU environments before it could be considered.

Lines 463 to 466: “However, bioaugmentation or biocontrol strategies remain highly preliminary and would require extensive safety assessments, long-term stability testing, and validation under real NICU conditions before being considered, particularly for vulnerable populations such as premature infants.”

Reviewer #2 (Comments for the Author):

Summary

This study provides a rigorous and multifaceted examination of the ecological determinants governing the persistence of *Pseudomonas aeruginosa*, *Serratia marcescens*, and *Stenotrophomonas maltophilia* in NICU sink drains. By combining longitudinal sampling, genotypic characterization, microbial community analysis, and abiotic measurements, the authors present one of the most comprehensive assessments to date of pathogenic colonization in hospital sink environments. A key strength of the work lies in its genotype-level resolution, which reveals low diversity and the long-term persistence of dominant sequence types within drains. This finding underscores the capacity of specific pathogenic lineages to stably colonize plumbing systems, potentially serving as chronic reservoirs for hospital-acquired infections. The study also demonstrates that spatial heterogeneity in pathogen distribution is driven primarily by variation in drain community composition, residual chlorine, and faucet design, illustrating how engineering and environmental parameters jointly shape microbial ecology in clinical settings. The isolation and characterization of *Delftia tsuruhatensis* Dt1S33 represent a notable contribution. Its negative correlation with opportunistic pathogens *in situ*, coupled with its inhibition of pathogen biofilm formation *in vitro*, provides compelling preliminary evidence for antagonistic interactions within sink microbiota. This insight suggests that ecological or probiotic strategies—such as bioaugmentation with antagonistic strains—may complement or enhance traditional disinfection approaches. However, the translational potential of such methods, particularly in vulnerable populations such as premature infants, requires careful evaluation. Overall, this study advances current understanding of environmental reservoirs of healthcare-associated pathogens and highlights the intertwined roles of biotic and abiotic factors in shaping sink colonization dynamics. The findings have important implications for infection control and point toward innovative, ecology-informed interventions capable of reducing pathogen persistence in high-risk hospital units.

Answer

We thank the reviewer for the thoughtful summary of our work and the positive feedback. We agree that ecological or probiotic strategies require careful and rigorous evaluation, particularly in vulnerable populations such as premature infants. This point is more emphasized in the revised manuscript (see Answer #12 for the first reviewer, and lines 463 to 466).

Re: mSystems01546-25R1 (Ecological dynamics of three persistent opportunistic pathogens in hospital sinks and their potential antagonistic bacteria)

Dear Dr. Emilie Bédard:

Your manuscript has been accepted, and I am forwarding it to the ASM production staff for publication. Your paper will first be checked to make sure all elements meet the technical requirements. ASM staff will contact you if anything needs to be revised before copyediting and production can begin. Otherwise, you will be notified when your proofs are ready to be viewed.

Sincerely,
Michaeline Albright
Editor
mSystems